# Comparison of Strengthening Solutions with Optimized Passive Energy Dissipation Systems in Symmetric Buildings

**Charbel Mrad** [1]**, Magdalini D. Titirla** [2,*]  **and Walid Larbi** [2]

[1] Department of Civil Engineering, Institut Supérieur des Sciences Appliquées et Économiques (ISSAE), Conservatoire National des Arts et Métiers (CNAM LIBAN), Beirut 20239201, Lebanon; charbel.mrad@isae.edu.lb

[2] Structural Mechanics and Coupled Systems Laboratory (LMSSC), Conservatoire National des Arts et Métiers (CNAM), 75003 Paris, France; walid.larbi@lecnam.net

[*] Correspondence: magdalini.titirla@lecnam.net

**Abstract:** The aim of this study is to compare the seismic response of reinforced concrete (RC) symmetric buildings, with a varied number of stories, strengthening with three types of passive energy dissipation systems, as tuned mass dampers, viscous dampers, and friction dampers. The paper presents an overview of design optimization with the object of minimizing certain functions: (i) the maximum displacement at the top of the structures, (ii) the base shear loads, and (iii) the maximum interstory drift. The objective functions were evaluated in three residents' buildings (a four-story building, a nine-story building, and a sixteen-story building) subjected to seven (real and artificial) seismic recorded accelerograms. For this purpose, 94 nonlinear dynamic analyses were carried out. The effects of each strengthening solution are presented, and from this innovative comparison (optimal design, three different passive energy systems, three different story numbers), further useful results were observed. The outcomes of the study show the effectiveness of a tuned mass damper (TMD) system, and how it might be better for tall and flexible structures than for stiffer structures. However, the response of the pendulum tuned mass damper (TMD) configuration is better than the conventional one because it acts in all directions. The viscous dampers (VDs) provide a significant reduction for mid-rise buildings, while friction dampers (FDs) boost the performance of all structures under seismic action, especially in terms of displacement, and they are more suitable for low-rise buildings.

**Keywords:** passive energy dissipation systems; tuned mass damper; viscous damper; friction damper; optimization; dynamic response

## 1. Introduction

During an earthquake, most structures have an inherent damping in them which results in some of the input seismic energy being dissipated, but a large amount of energy is absorbed by the structure, causing it to undergo several deformations and maybe even collapse. So, over the last year, there has been great interest in the creation of seismic energy dissipation devices that will absorb the majority of the seismic energy, but will not belong to the supporting structure of the construction (conventional braced frames). The main advantages of these are their easy replacement or repair. These devices belong to the passive energy dissipation systems, do not require external power to generate system control forces, and hence, are easy and cheap to implement in a structure [1–3]. Passive energy dissipation devices such as tuned mass dampers (TMD), viscous dampers (VD), and friction dampers (FD) have widely been used to reduce the dynamic response of civil engineering structures that are subjected to seismic loads. Their effectiveness for the seismic design of building structures is attributed to minimizing structural damages by absorbing the structural vibratory energy and by dissipating it through their inherent hysteresis behavior.

The passive TMD is undoubtedly a simple, inexpensive, and somewhat reliable means to suppress the undesired vibrations. The TMD concept was first applied by Frahm in 1909 [4] to reduce the rolling motion of ships and ship hull vibrations. A theory for the TMD was presented later by Ormondroyd and Den Hartog [5], followed by a detailed discussion of optimal tuning and damping parameters in Den Hartog's book on mechanical vibrations [6]. A number of TMDs have been installed in tall buildings, bridges, and towers. The first structure in which a TMD was installed is the Centrepoint Tower in Sydney Australia, which was conceived in 1968 [7]. There are many buildings in the United States, like the Citicorp Center in New York City [8] and the John Hancock Tower in Boston [9]; in Japan, there is the Chiba Port Tower [10] and others [11].

A VD damper generally consists of a piston within a damper housing filled with a compound of silicone or a similar type of oil, with the piston containing a number of small orifices through which the fluid may pass from one side of the piston to the other [12]. Viscoelastic materials are very popular in engineering [13,14]. As the damper piston rod and piston head are stroked, fluid is forced to flow through orifices either around or through the piston head. The first applications of VD dampers to structures were for reducing acceleration levels, or increasing human comfort, due to wind. In 1969, VD dampers were installed in the twin towers of the World Trade Center in New York as an integral part of the structural system. In 1982, VD dampers were incorporated into the 76-story Columbia Sea First Building in Seattle, Washington, to protect against wind-induced vibrations [15]. Applying the well-developed fluid damping technology to civil structures was relatively straightforward; within a short time, the first research projects were completed on the application of fluid dampers to a steel framed building [12] and an isolated bridge structure [16].

In a typical FD, the generated frictional force helps to dissipate the external energy and stabilize the structure under the dynamic excitation scenarios. The FDs are also not prone to thermal effects, and possess a stable hysteretic behavior for a considerable number of cycles under such dynamic excitations [17]. Based primarily upon an analogy to the automotive brake, Pall et al. [18] began the development of passive frictional dampers to improve the seismic response of structures. The objective is to slow the motion of buildings "by braking rather than breaking" [19]. After that, many researchers proposed friction dampers that focus on protection in the braced frames or in the joint connection [20–27]. Several of these devices have been selected for the seismic strengthening of existing or new buildings in the USA, Canada, and Japan [28–30].

This study compares the seismic response of three reinforced concrete (RC) symmetric buildings of varying stories and their strengthening with three types of passive energy dissipation systems, as tuned mass dampers, viscous dampers, and friction dampers. We focus on the optimal design of each building in minimizing (i) the maximum displacement at the top of the structures, (ii) the base shear loads, and (iii) the maximum inter-story drift. Three residents' buildings (a four-story building, a nine-story building, and a sixteen-story building) were subjected to seven (real and artificial) seismic recorded accelerograms. For this purpose, 94 nonlinear dynamic analyses were carried out. The effects of each strengthening solution are presented, and from this innovative comparison (optimal design, three different passive energy systems, and three different story numbers), further useful results were observed.

## 2. Description of Benchmark Investigated Buildings

Three symmetric, in plan, reinforced concrete residential buildings were studied in this paper. The three buildings were regular in plan according to EC8 [31], and they had the same external dimensions: 40.00 m in the longitudinal direction and 20.00 m in the transversal direction, as shown in Figure 1. The number of the stories was varied, with the constant height of each story equal to 3.50 m. The first building, mentioned from now on as « Low-rise », consisted of three stories; the second building, mentioned as « Mid-rise », consisted of eight stories; the third building, mentioned as « High-rise », consisted of 15

stories. More details of the construction elements are given in Table 1. The buildings had a structural system for resisting horizontal loads based to walls. Their distribution in plan was symmetric in both horizontal directions to avoid an additional torsional effect.

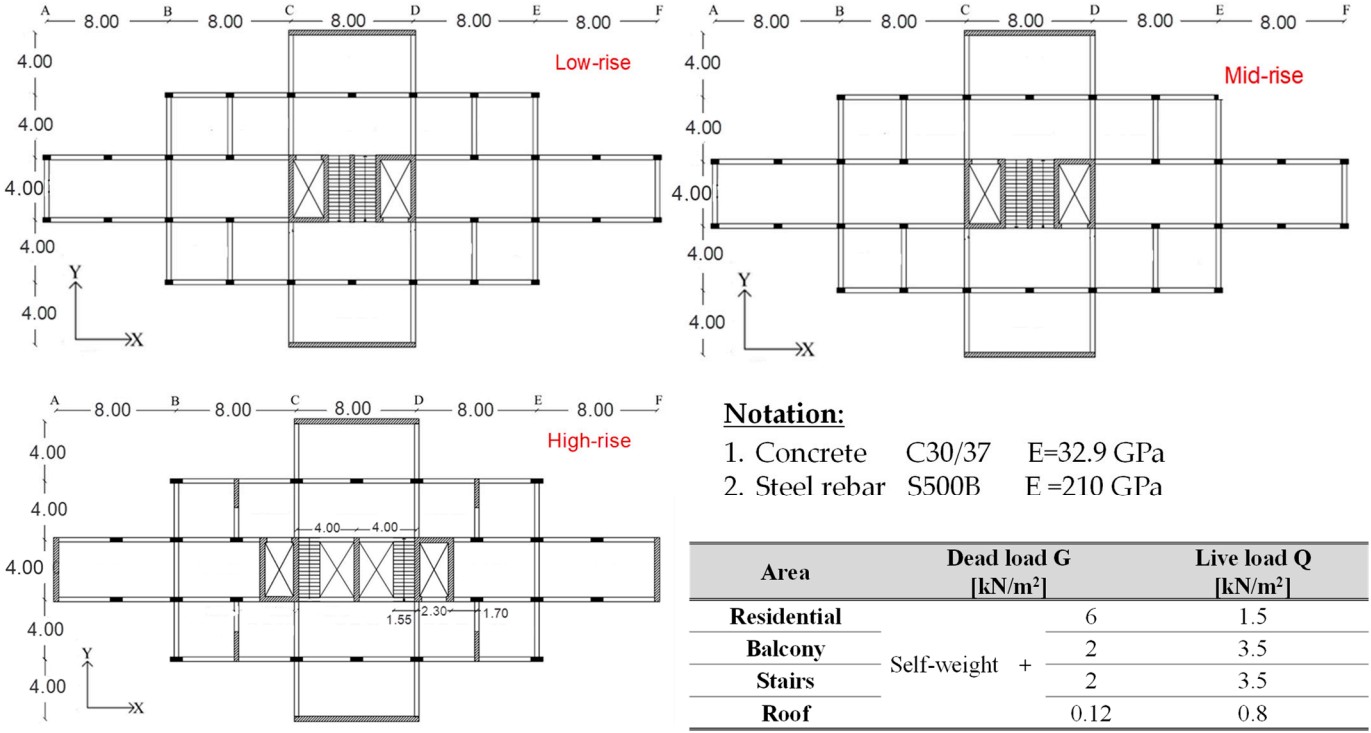

**Figure 1.** Plan view of the buildings (units in m).

**Table 1.** Description of the investigated buildings.

| Building Type | Low-Rise | Mid-Rise | High-Rise |
|---|---|---|---|
| Number of stories | G + 03 | G + 08 | G + 15 |
| Story height | | 3.5 m | |
| Total height including roof level | 14 m | 31.5 m | 56 m |
| Columns cross section | 30 × 50 cm | 30 × 80 cm | 30 × 80 cm |
| Beams cross section | | 30 × 30 cm | |
| Walls thickness | | 30 cm | |
| Slab thickness | | 20 cm | |

## 3. Building's Modeling

A finite element method (FEM) was constructed to model the structural system and mass distribution. Non-linear dynamic time history analyses were performed to account for geometrical and structural non-linearities. The beams and the columns were modeled as frame elements with rectangular cross sections (see Table 1), while the walls were modeled as shell elements. The rigid floor diaphragm assumption was used for the modeling of the stories, as the buildings are regular (EN 1998-1:2004, page 42, section 4.2.3.2) and in elevation (EN 1998-1:2004, page 43, section 4.2.3.3) [31]. For the walls and floors, a four-node shell elements was used in this study. The selected shells elements are homogeneous, with 6DOF in each node, and an appropriately selected mesh was used in order to have equilibrium between the accuracy of the results and the computation cost (103,200 DOF for the « low-rise » building, 272,400 for the « mid-rise », and 628,800 for the « high-rise »). In the three-dimensional structural model, elastic flexural stiffness and shear stiffness were taken into account, and equal to the one-half of the corresponding stiffness of the uncracked



elements [31]. Material properties like concrete and steel rebars remain the same for all the stories, while the building is subjected to gravity and lateral loads (see Figure 1).

The optimal design of the strengthening solutions is presented in the Section 4 of this paper. A tuned-mass damper (TMD), also known as a pendulum damper, is not actually a damper, but rather a pendulum or another gravity-based oscillator that is attached to the structure in such a way that it counteracts the vibration of one or more fundamental modes, thereby reducing the wind and/or seismic response of those modes. A TMD was modeled using a spring-mass system with damping. A linear link element reproduced the spring properties, while the mass and weight was also assigned in the model. The details of the TMD mass (or PTMD) for each building are presented in the Section 4.1. The damping properties of nonlinear viscous dampers (VD) were based on the Maxwell model of viscoelasticity [32]. The nonlinear properties, as stiffness, damping coefficient, and damping exponent were specified, and modeled in series. A linear link object is most suitable unless nonlinear damping is assigned using a damping exponent other than 1.0. This enables the modeling of a linear dashpot parallel with linear stiffness for both linear and nonlinear analysis cases. The numerical modeling of friction dampers (FD) was very easy, since the hysteretic loop of the friction dampers is perfectly rectangular, similar to the perfectly elasto-plastic material. The friction dampers were modeled as a fictitious plasticity element having a yield force equal to the slip load. The FD and VD were positioned in steel diagonal brace elements. More details for the shape and the position of the steel diagonal braces are given in the Section 4 of this paper. The braces were modeled as a frame element.

All building models, i.e., the benchmark buildings and the alternative ones with the passive energy dissipation systems, were analyzed for seven different real and artificial accelerograms that were compatible to ground type B-dependent Eurocode 8 elastic spectra (seismic zone V according to the French national annex [33]). The selection of the accelerograms was based on the provisions of Eurocode 8 Part 1 [31]. The mass and stiffness proportional damping was chosen, and critical damping ratios equal to 5% and 4% were considered for the first and second period of the analyzed building systems, correspondingly. The strengthening solutions were carried out by maximizing structure performances as much as possible. This can be done by adopting an elastic linear behavior (behavior factor $q$ = 1) to help prevent damages in structural elements that could compromise the durability of the structures. In order to take into account uncertainties linked to the location of the masses and the spatial variation of the seismic movement, EC8 requires an additional accidental eccentricity of at least 5% of the dimension of the building that is perpendicular to the direction of seismic action. This means that the center of gravity of each story must be offset in each direction of this eccentricity with respect to its nominal position. The adopted 5% eccentricity is considered constant, and repeated on each story in the same direction in the present study.

A nonlinear dynamic analysis was performed with seismic inputs described by bi-directional recorded accelerograms (Figure 2a), which have been applied at base level. Figure 2b shows the response spectra of the selected accelerograms compared to the EC8 elastic response spectrum Type 1, with peak ground acceleration equal to 0.3 g, ground type B, and 5% damping.

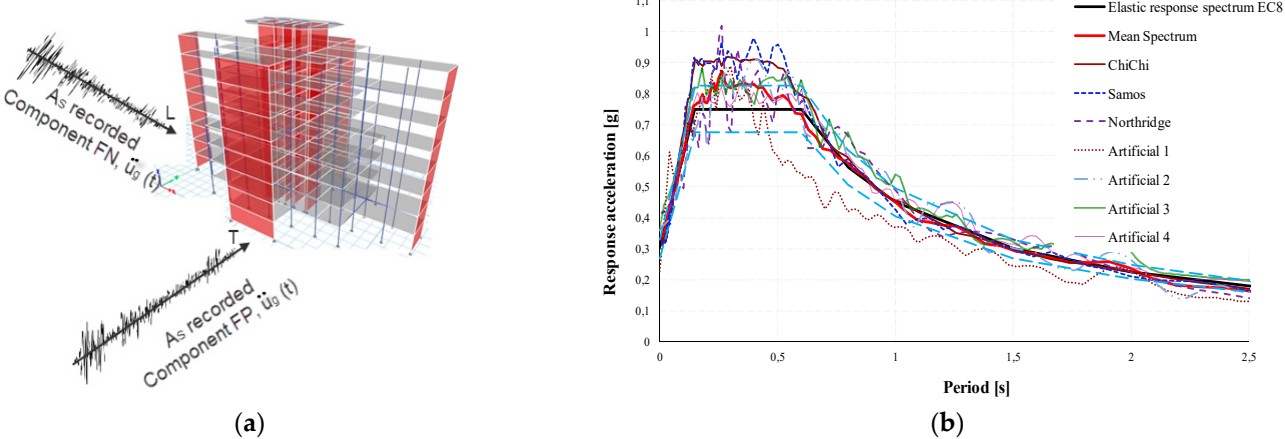

**Figure 2.** (**a**) The bidirectional recorded accelerograms, (**b**) response spectra relative to the selected accelerograms compared to the EC8 elastic response spectrum Type 1 with peak ground acceleration equal to 0.3 g, ground type B, and 5% damping.

## 4. Optimal Design of Passive Energy Dissipation Systems

These days, there are numerous passive energy dissipation dampers, while the present study is focused on TMD, VD, and FD, which are described in the introduction. A design optimization technique for each system is presented in this section, covering damper characteristics and displacements. The design optimization sought to minimize (i) the maximum displacement at the top of the structures and (ii) the maximum inter-story drift.

### 4.1. Tuned Mass Damper

TMD is a motion-based passive system that consists of a mass $m_d$, a spring with spring stiffness $k_d$, and a dashpot with a damping coefficient $c_d$ attached and typically tuned to the natural structural frequency [34]. During an earthquake, the damping system is stretched and compressed, reducing vibrations in the structure by increasing its effective damping [34]. A schematic representation of the 2 DOF (degree of freedom) system is shown in Figure 3, noting that $m$, $k$, and $c$ represent, respectively, the main mass, stiffness, and inherent damping coefficient of the structural system. TMD is typically effective over a narrow frequency band. It is therefore important to be tuned to a particular natural frequency. The system efficiency decreases, with structures having several closely spaced natural frequencies [34].

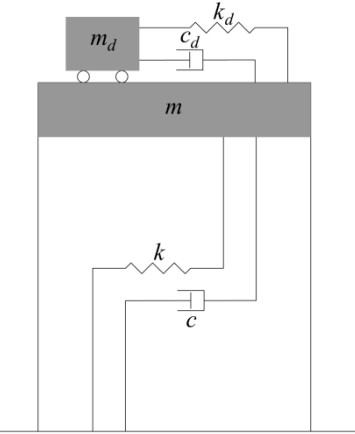

**Figure 3.** Schematic representation of the 2 DOF system.

An appreciation of TMD efficiency can be obtained by following the basic development of Den Hartog [6], which considers an undamped structural system subject to a sinusoidal excitation. Figure 4 shows that the dynamic amplification factor, *R*, which takes the

damping effect of the TMD, is a function of the four essential variables: the mass ratio $\overline{m}$ ($\overline{m} = m_d/m$), the TMD damping ratio $\xi_d$ ($\xi_d = \frac{c_d}{2 \cdot m_d \cdot \omega_d}$), the frequency ratio $\nu$ ($\nu = \omega_d/\omega$), and the forced frequency ratio $\lambda$ ($\lambda = \overline{\omega}/\omega$), where $\omega = \sqrt{k/m}$ and $\omega_d = \sqrt{k_d/m_d}$ are the natural frequency of the structural system and TMD, respectively. The dynamic amplification factor $R$ is expressed by the Equation (1):

$$R = \sqrt{\frac{(\nu^2 - \lambda^2)^2 + (2 \cdot \xi_d \cdot \nu \cdot \lambda)^2}{[(\nu^2 - \lambda^2)(1 - \lambda^2) - \nu^2 \cdot \lambda^2 \cdot \overline{m}]^2 + (2 \cdot \xi_d \cdot \nu \cdot \lambda)^2 (1 - \lambda^2 - \lambda^2 \cdot \overline{m})^2}} \tag{1}$$

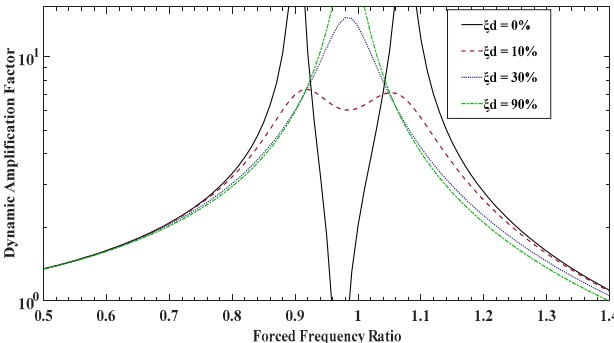

**Figure 4.** Dynamic amplification factor as a function of $\lambda$.

Figure 4 shows a plot of $R$ as a function of $\lambda$ for $\overline{m} = 0.05$ and $\nu = 1$. Without TMD damping, the response amplitude is infinite at two resonant frequencies of the 2 DOF systems. Furthermore, for an infinite TMD damping, the two masses are virtually fused to each other, leading the amplitude of resonant frequency to be infinite again [1]. Therefore, between these extremes, there is a value of $\xi_d$, for which the peak becomes a minimum.

An objective of installing TMD in structures is to bring the response amplitude down to its lowest possible value, 1; this is why the damping ratio of TMD must be carefully selected in such a way that small amplifications over a wider frequency bandwidth can be achieved. As can be seen in Figure 4, this can be achieved by taking a small value of $\xi_d$, like 30%. So, the effect of the TMD damping ratio is very essential. One observes that this parameter must exist but must not be high, because at this case, the amplifications are small and the frequency range in which the damper works is the biggest increasing damper efficiency. Outside of this range, the motion is not considerably influenced by the TMD system.

The conventional TMD described above requires a large mass and space for installation, thus creating architectural constraints [35]. An alternative approach is using a pendulum configuration PTMD. During ground motion, the pendulum produces a horizontal force which opposes the story motion [36]. This configuration type can be represented by an equivalent SDOF system attached to the story, as shown in Figure 5.

With the pendulum configuration [36], the equivalent stiffness is given by $k_{eq} = m_d \cdot g/L$, the natural frequency is expressed by $\omega_d = \sqrt{k_{eq}/m_d} = \sqrt{g/L}$ and the natural period is set by $T_d = 2 \cdot \pi \cdot \sqrt{L/g}$. The tuning parameters of PTMD are the mass $m_d$ and the length $L$. This configuration is advantageous over the conventional TMD, especially for high-rise buildings, because its frequency can be retuned easily by modifying the cable length [37].

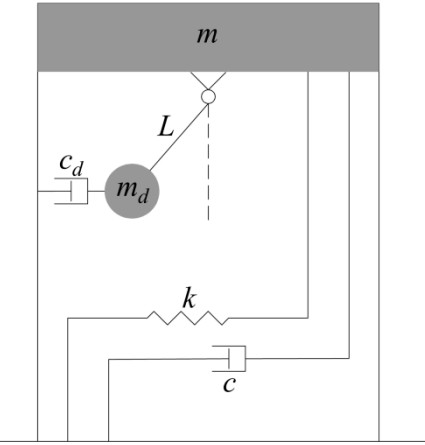

**Figure 5.** PTMD schematic representation.

The first important parameter in the optimal design process of the TMD system is the mass ratio $\overline{m}$. It is well known that the structural response decreases as $\overline{m}$ increases, but this ratio has a limit in practice which must not exceed 10% [34] based on geometrical and economical constraints. Then, optimized absorber parameters are calculated.

All real systems contain some inherent damping, meaning that an absorber is added to a lightly damped system. The effect of the inherent damping in the real system is an important design consideration on the optimum tuning parameters of TMD. Table 2 shows the equations used in the present study for $v$ and $\xi_d$, which includes the structural damping ratio of the primary structural system $\xi$, the derived rigidity $k_d$, and damping coefficient $c_d$ of the TMD, respectively [38]. Also, Table 3 shows the structure and TMD optimal parameter values taken for the three investigated buildings, noting that $T$ is the fundamental period of the structure in the transversal direction in which the TMD is applied.

**Table 2.** TMD optimized parameters expressions.

| Expressions | | |
|---|---|---|
| Optimal tuning parameters of TMD given in [38] | $v = \frac{1}{1+\overline{m}} \cdot \left[1 - \xi \cdot \sqrt{\frac{\overline{m}}{1+\overline{m}}}\right]$ | (2) |
| | $\xi_d = \frac{\xi}{1+\overline{m}} + \sqrt{\frac{\overline{m}}{1+\overline{m}}}$ | (3) |
| Optimized absorber parameter | $k_d = \omega_d^2 \cdot m_d = v^2 \cdot \omega^2 \cdot \overline{m} \cdot m$ | (4) |
| | $c_d = 2 \cdot \xi_d \cdot \omega_d \cdot m_d = 2 \cdot \xi_d \cdot v \cdot \omega \cdot \overline{m} \cdot m$ | (5) |

**Table 3.** Structure-TMD optimal parameter values.

| Building Type | Structure | | TMD | | |
|---|---|---|---|---|---|
| | Parameter | Value | Parameter | Values | |
| Low-rise | $T$ [s] | 0.216 | $\overline{m} = 0.5\%$ | $v = 0.9915$ | $\xi_d = 12.03\%$ |
| | $m$ [t] | 1572.29 | $\overline{m} = 1\%$ | $v = 0.9852$ | $\xi_d = 14.90\%$ |
| Mid-rise | $T$ [s] | 0.985 | $\overline{m} = 1\%$ | $v = 0.9852$ | $\xi_d = 14.90\%$ |
| | $m$ [t] | 4626.92 | $\overline{m} = 3\%$ | $v = 0.9626$ | $\xi_d = 21.92\%$ |
| High-rise | $T$ [s] | 2.202 | $\overline{m} = 2\%$ | $v = 0.9735$ | $\xi_d = 18.90\%$ |
| | $m$ [t] | 8560.39 | $\overline{m} = 3\%$ | $v = 0.9626$ | $\xi_d = 21.92\%$ |

In order to choose the most appropriate mass ratio, two analyses will be carried out for two mass ratios per building (Table 4). The structural response will be compared to the undamped case in Table 5.

**Table 4.** TMD design parameter values for the three investigated buildings.

| Building Type | $\bar{m}$ | $m_d$ [t] | $k_d$ [kN/m] | $c_d$ [kN·s/m] |
|---|---|---|---|---|
| Low-rise | 0.50% | 7.86 | 6561.47 | 54.63 |
| | 1% | 15.72 | 12,955.58 | 134.49 |
| Mid-rise | 1% | 46.27 | 1829.10 | 86.70 |
| | 3% | 138.8 | 5238.61 | 373.86 |
| High-rise | 2% | 171.2 | 1321.15 | 179.82 |
| | 3% | 256.8 | 1937.40 | 309.24 |

**Table 5.** Responses of the three investigated buildings for the two different mass ratio of TMD with margin from undamped case.

| Building Type | Case | Direction | Fundamental Period | | Top Roof Displacement | | Base Shear | |
|---|---|---|---|---|---|---|---|---|
| | | | Value [s] | Margin | Value [cm] | Margin | Value [kN] | Margin |
| Low-rise | Undamped | Longitudinal | 0.156 | | 0.8556 | | 9536.64 | |
| | | Transversal | 0.216 | | 0.4248 | | 2772.68 | |
| | Damped with $\bar{m} = 0.5\%$ | Longitudinal | 0.158 | −1.28% | 0.8243 | 3.66% | 9530.28 | 0.07% |
| | | Transversal | 0.218 | −0.93% | 0.4183 | 1.53% | 2751.67 | 0.76% |
| | Damped with $\bar{m} = 1\%$ | Longitudinal | 0.157 | −0.64% | 0.8237 | 3.73% | 9287.71 | 2.61% |
| | | Transversal | 0.216 | 0.00% | 0.4164 | 1.98% | 2693.44 | 2.86% |
| Mid-rise | Undamped | Longitudinal | 0.697 | | 13.2068 | | 24,870.8 | |
| | | Transversal | 0.985 | | 5.4141 | | 4652.36 | |
| | Damped with $\bar{m} = 1\%$ | Longitudinal | 0.71 | −1.87% | 12.6213 | 4.43% | 23,726.16 | 4.60% |
| | | Transversal | 0.997 | −1.22% | 5.3094 | 1.93% | 4567.02 | 1.83% |
| | Damped with $\bar{m} = 3\%$ | Longitudinal | 0.733 | −5.16% | 12.5036 | 5.32% | 22,507.32 | 9.50% |
| | | Transversal | 1.031 | −4.67% | 4.7931 | 11.47% | 4559.55 | 1.99% |
| High-rise | Undamped | Longitudinal | 1.983 | | 38.377 | | 19,306.7 | |
| | | Transversal | 2.202 | | 12.5677 | | 5222.92 | |
| | Damped with $\bar{m} = 2\%$ | Longitudinal | 2.058 | −3.78% | 35.9845 | 6.23% | 18,077.8 | 6.37% |
| | | Transversal | 2.284 | −3.72% | 10.2574 | 18.38% | 5112.19 | 2.12% |
| | Damped with $\bar{m} = 3\%$ | Longitudinal | 2.091 | −5.45% | 36.3981 | 5.16% | 17,476.06 | 9.48% |
| | | Transversal | 2.322 | −5.45% | 10.1373 | 19.34% | 5331.14 | −2.07% |

Flexible buildings undergo larger horizontal displacements, which may result in significant damages. In this case, it is preferable to choose a relatively big mass ratio, unlike rigid buildings, for which it is advisable to adopt a relatively small ratio, because increasing its value does not provide any additional damping effect.

This is why the low-rise building is studied over the two following small mass ratio values 0.5% and 1%, the mid-rise building over 1% and 3%, while the high rise building will be studied for 2% and 3%.

As for the installation location of TMD, choosing the best location is the most important factor to consider in the optimal design process to show excellent control performance for the controlling dynamic response [39]. It is important to note that there are not enough studies carried on the installation of TMD in a spatial structure, as well as a lack of data on the optimal installation [40].

The TMD is commonly installed at the center plan to avoid creating torsional effects. In their study on the performance and placement of one or more TMDs in buildings, Almazan et al. [41] concluded that the optimum location is near the geometric center of the plan, whether for symmetric or asymmetric buildings. In addition to that, TMD is a motion-based system, which means that the TMD efficiency in reducing structural response is gained by applying it at the story that will experience the most motion. In symmetric buildings, it is usually on the upper story level [42].

The margins in Table 5 are calculated according to the undamped case, which means that a reduction is detected for positive values. The negative margins mean that there is an increase in parameter values after installing the TMD.

After focusing on the margin between the two proposed mass ratio for each of the three investigated buildings, it is clear that an increase in the mass ratio brings a relatively small additional damping for Artificial 4 ground motion. This is why for economic concerns, 0.5% is adopted for the low-rise building as a value of mass ratio, and 3% for the mid and high-rise building.

Optimized TMD design parameter values are listed in Table 6. However, the optimal position for irregular buildings is not necessarily on the upper story due to the different stiffness values for each story in elevation. Furthermore, the damper location in plan is considered a primary design variable; in this case, it depends essentially on the eccentricity between the center of mass and rigidity, but is always near the geometric center of the plan [41]. It is important to note that buildings optimal design is evaluated for Artificial 4 ground acceleration by applying a nonlinear time history analysis because its spectral response is close to the EC8 elastic response spectrum [31].

**Table 6.** Cable length required for the three investigated buildings.

| Building Type | $\bar{m}$ | $\omega_d$ [rad/s] | $L$ [m] |
| --- | --- | --- | --- |
| Low-rise | 0.50% | 28.893 | 0.012 |
| Mid-rise | 3.00% | 6.143 | 0.259 |
| High-rise | 3.00% | 2.747 | 1.300 |

An alternative solution is using a PTMD configuration, which can provide additional damping, as it can act in all directions. Table 6 shows that the cable length required for the low and mid-rise building are too small, and thus, unrealistically small values. So, the PTMD configuration is only evaluated for the high-rise building. The cable that relates the additional mass to structure is composed of steel rods, with a circular section offering high axial rigidity. In this optimization section, a comparison between TMD and PTMD configuration is established. It is clear that the values obtained for the three parameters studied are relatively close. This can be explained by the fact that the building is not too flexible, which limits the performance of TMD and PTMD as well (Figure 6). However, the PTMD configuration is adopted in the high-rise building because it offers more reduction than the translational configuration, and its frequency can be easily retuned (Table 7).

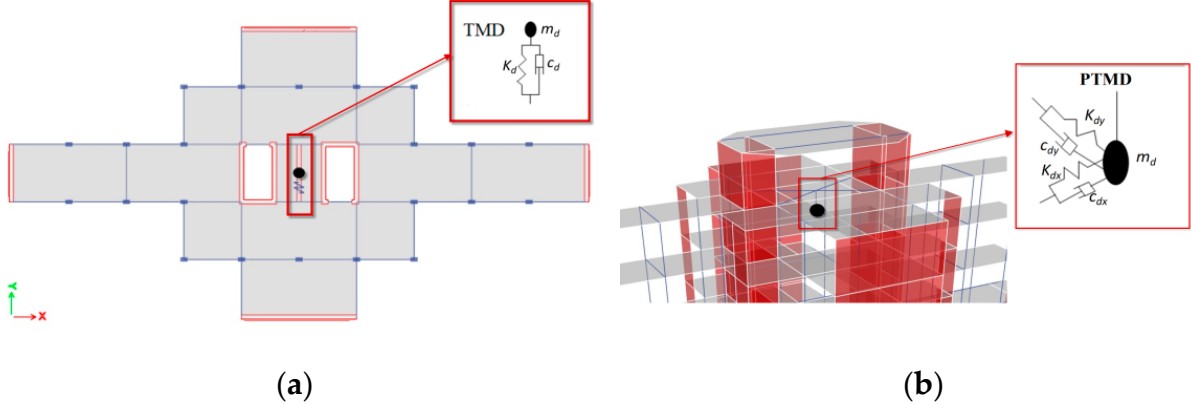

**(a)**        **(b)**

**Figure 6.** Modeling configuration of (**a**) TMD for low-rise and mid-rise buildings, (**b**) PTMD for high-rise building.

**Table 7.** Response of the high-rise building for the TMD and PTMD configuration.

| Case | Longitudinal | | | Transversal | | |
|---|---|---|---|---|---|---|
| | Undamped | With TMD | With PTMD | Undamped | With TMD | With PTMD |
| Fundamental period [s] | 1.983 | 2.091 | 1.801 | 2.202 | 2.322 | 2.67 |
| Top roof displacement [cm] | 38.377 | 36.39 | 25.01 | 12.567 | 10.14 | 9.23 |
| Base shear [kN] | 19,306.7 | 17,476.06 | 16,974.11 | 5222.9 | 5331.14 | 4386.71 |

*4.2. Viscous Dampers (VDs)*

Viscous damping is the dissipation of energy that occurs when a particle in a vibrating system is resisted by a force, the magnitude of which is a constant independent of displacement and velocity, and the direction of which is opposite to the direction of the velocity of the particle. Uniaxial force is a result of a pressure across the piston head. Since the fluid is nearly incompressible, a reduction in fluid volume results in a restoring force, which is prevented by the use of a rod make-up accumulator [12]. Previous research show an increase in temperature can be significant, particularly in long-durations or large-amplitude seismic motions. This temperature is compensated by a relatively small effect through mechanisms [43,44].

An efficient mathematical model to describe VD behavior (linear or nonlinear) was proposed by Seleemah and Constatinou [45] based on experimental results. The force of the damper $P(t)$ is calculated by the following Equation (6):

$$P(t) = C_d |\dot{u}(t)|^\alpha \, sgn[\dot{u}(t)] \tag{6}$$

where $C_d$ is the damping coefficient, $u(t)$ is the displacement across the damper, and $\alpha$ is a coefficient, depending on the piston head design and viscosity properties of fluid.

The coefficient $\alpha$ is the first important parameter to verify which could be less or equal to 1. Figure 7a describes the force-velocity relationship for linear and nonlinear behavior, while Figure 7b shows the force–displacement hysteretic loops. For earthquake resistance structures, the $\alpha$ coefficient has a value ranging from 0.3 to 1.0, in order to provide larger forces and to minimize shocks for high velocities with no degradation of performance [46]. In addition, the lowest value needed to maintain a high amount of energy absorbed per cycle of vibration is shown in Figure 7b, and minimizes at the same time the stress at adjacent structural members [47]. So, in our study, an $\alpha$ value equal to 0.3 has been selected.

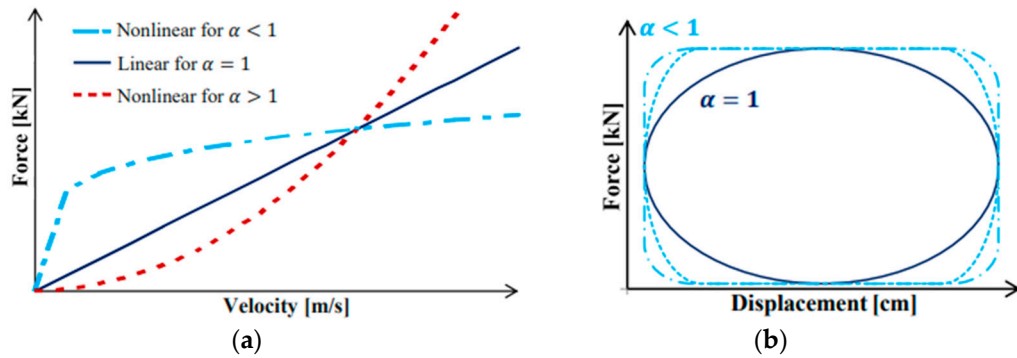

(a)　　　　　　　　　　　　　(b)

**Figure 7.** (**a**) Force-velocity behavior of VD and (**b**) Force-displacement hysteretic-loops.

Velocity is the second important parameter to fix. This is because the VD force varies with velocity, which is related to structural motion and depends on the structural fundamental period. The horizontal flexibility of the structure injects the full movement directly into the horizontal component of the damper, so VD is considered efficient for flexible rather than rigid structures [47]. In this study, a parametric study was done in order to select the correct velocity in accordance with previous studies [47].

The damping coefficient (*Cd*) is the third important parameter to define, related to the desired effective damping $\xi_{eff}$, and attributed to the structure. Design codes do not provide any substantial procedure for the distribution of the calculated damping coefficient over the whole building. These days, a large variety of methods have been proposed, classified between two categories: standard and advanced methods [48,49]. In the present study, the damping coefficient is distributed along the height of the building, based on the proportionality respective of the story shear force (Equation (7)); the effective damping $\xi_{eff}$ is a sum of the structural inherent damping ratio ($\xi_0$) and the damping ratio of the viscous dampers ($\xi_d$), according to [50] recommendations (see Equation (8)).

$$C_{d,i} = \frac{V_i}{\sum V_i} \sum C_j \tag{7}$$

$$\xi_{eff} = \xi_0 + \xi_d = \xi_0 + \frac{\sum \lambda C_j \phi_{rj}^{1+\alpha} \cos^{1+\alpha} \theta_j}{2\pi A^{1-\alpha} \omega^{2-\alpha} \sum M_i \phi_i^2} \tag{8}$$

$$\lambda = 2^{2+\alpha} \frac{\Gamma^2 \left(1 + \frac{\alpha}{2}\right)}{\Gamma(1+\alpha)} \tag{9}$$

where *A* is the amplitude, $\phi_{rj}$ is the relative horizontal displacement of the damper, $\theta_j$ is the inclined angle of the damper *j*, $\omega$ is the loading frequency supposed equal to the natural structural frequency, $M_i$ is the vibrating mass of the story *i*, $\phi_i$ is the modal displacement at story i, and $\lambda$ is a parameter calculated by Equation (9) [51].

Del Gobbo [52] indicates that in order to establish the optimal effective damping, nonstructural elements must be taken into account. To have an essential damping ratio-repair cost relationship, the range of optimal effective damping is identified as $30 - 40\%$ to minimize mean economic losses. However, the optimal damping amount also depends on the building's properties, such as the fundamental period of structure. Table 8 shows the selected, effective damping and velocities values, as well as the calculated damping coefficient.

**Table 8.** Effective damping and calculated damping coefficient for the three investigated buildings ($\alpha = 0.3$).

| Building Type | Direction | Fundamental Period [s] | Structural Rigidity Description | Suggested Velocity [m/s] | Suggestedeffective Damping $\xi_{eff}$ | $\sum C_j$ [kN·(s/m)] |
|---|---|---|---|---|---|---|
| Low-rise | Longitudinal | 0.156 | Rigid | 0.127 | 30% | 71,537.07 |
| | Transversal | 0.216 | | 0.127 | 30% | 59,405.58 |
| Mid-rise | Longitudinal | 0.697 | Semi-rigid | 0.254 | 35% | 56,981.16 |
| | Transversal | 0.985 | | 0.254 | 35% | 39,325.16 |
| High-rise | Longitudinal | 1.983 | Flexible | 0.381 | 40% | 12,658.71 |
| | Transversal | 2.202 | | 0.381 | 40% | 12,095.19 |

The design optimization of VD is not limited only on the mechanical parameters of VDs, but also on the position of the dampers in the plan-view of the building. It is important to ensure that the dampers are located in a configuration that does not introduce eccentricity to the structure; this is why the most efficient placement would be equivalently about the building's center of mass, for example, along the perimeter of typical structures. The main reason is to be able to control any torsional motion of the building [47]. In our study, the VDs are positioned in steel diagonal braces, half of them working under compression, and the other under tension. Different configurations of VD's placement were studied, while two of them are illustrated in the Figure 8. At least two dampers were positioned in each direction and on each side of the building's center mass at every story, even though it is not required. It could be terminated before the top levels or alternated at different story levels. Moreover, to limit damper force output, more than two dampers per direction could be used, especially for buildings with large footprints.

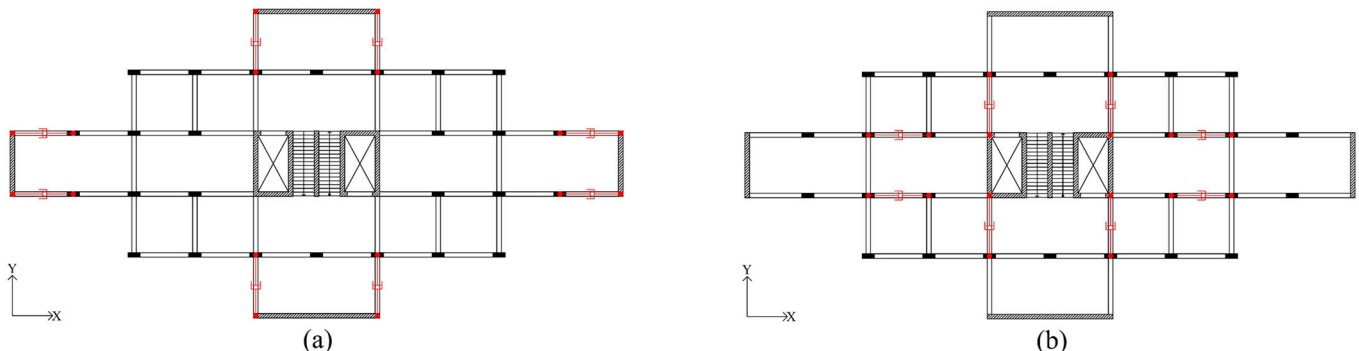

**Figure 8.** (**a**) Alternative 1 of dampers placement and (**b**) Alternative 2 of dampers placement.

To choose the best one for each building by evaluating the fundamental period, the top roof displacement and base shear obtained in the longitudinal and transversal directions is indicated in Table 9. It should be noted that dampers were installed at all levels with four systems per direction. It must be noticed that the dampers placement can affect a building's structural response by evaluating the structural response between the two alternatives of dampers placement shown in Table 9. In general, alternative 2 provides the best reduction, especially for the low and mid-rise building, and alternative 1 offers the smallest values of displacement for the high-rise building. So, alternative 2 is chosen for the low and mid-rise building, and alternative 1 for the high-rise building. Figure 9 shows the schematic configuration during the modeling of VDs.

**Table 9.** Responses of the three investigated buildings for the two alternatives of VDs placement.

| Building Type | Direction | Fundamental Period [s] | | Top Roof Displacement [cm] | | Base Shear [kN] | |
|---|---|---|---|---|---|---|---|
| | | Altern. 1 | Altern. 2 | Altern. 1 | Altern. 2 | Altern. 1 | Altern. 2 |
| Low-rise | Longitudinal | 0.112 | 0.112 | 0.167 | 0.176 | 1973.2 | 1905.6 |
| | Transversal | 0.148 | 0.147 | 0.218 | 0.345 | 2308.6 | 2070.0 |
| Mid-rise | Longitudinal | 0.387 | 0.387 | 3.121 | 3.763 | 1888.1 | 7567.0 |
| | Transversal | 0.775 | 0.775 | 2.997 | 1.760 | 4991.5 | 83.3 |
| High-rise | Longitudinal | 0.543 | 0.543 | 5.812 | 6.821 | 10,828 | 322.5 |
| | Transversal | 0.989 | 0.989 | 4.604 | 4.803 | 178.6 | 14,737.6 |

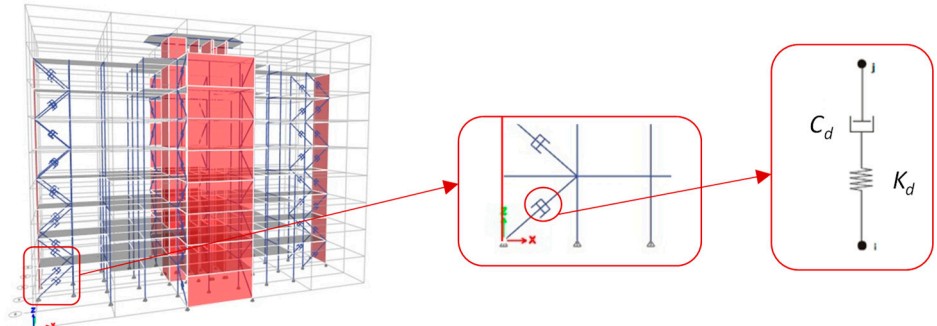

**Figure 9.** Modeling configuration of VDs.

### 4.3. Friction Dampers (FDs)

FD is a displacement-based system which dissipates energy through friction across the surfaces between two solid elements [1,3]. The dissipative mechanism generates heat through dry sliding friction with a stable hysteretic behavior [53]. A simple model for defining the behavior of the damper is given by the idealized Coulomb model of friction. The theory is based on the following hypotheses, which are experimentally validated [19]:

- Force independent of the apparent contact surface

- Force proportional to the total normal force acting through the interface
- Force independent of speed even with a slip at low speed

As a result, the force can be written using the following expression $F_t = \mu \cdot F_n$, where $F_t$ and $F_n$ represent the frictional and normal forces, respectively, and $\mu$ the coefficient of friction which depends on the selection of sliding materials and present conditions of the sliding interface. $F_n$ and $\mu$ are maintained at constant values over extended durations of time, which is difficult to achieve in practice [3]. The damper hysteresis loop is rectangular, showing a great amount of energy dissipated per cycle of motion, and the cyclic behavior of FD is strongly nonlinear, as shown in Figure 10a. When the friction force is overcome, FD adds initial stiffness to the structural system. It is important to note that if no restoring force is provided, permanent structural deformation may exist after an earthquake [3]. As shown in Figure 10b, the response of the structure is highly affected by FD slip force, and a small variation of FD optimum slip load has a minimum effect on structure's response. The selected slip force must be high enough to prevent the damper from slipping under a small applied lateral loads value, and should be low enough to achieve slip before the yielding of the main structural elements [54].

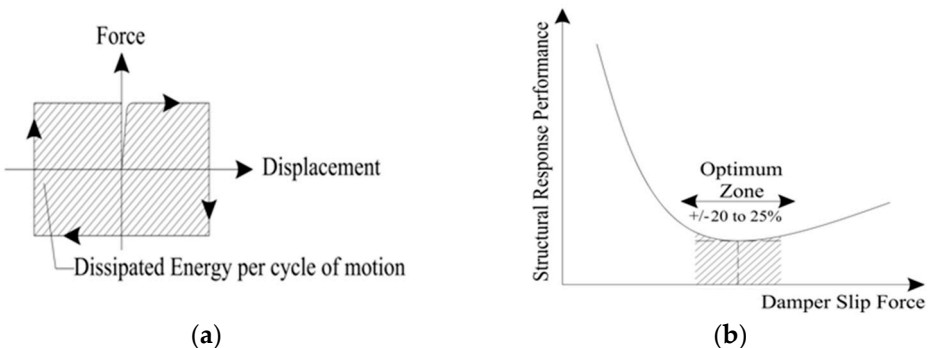

**Figure 10.** (**a**) Force-Displacement hysteresis loop of a friction damper, (**b**) optimal slip force effect on structural response.

A simple method used in the present study consists of taking a portion from the applied shear force, so the load at each story is estimated by the Equation (10)

$$F_{t,\, optimal} = \frac{1}{3}[V_i / n_i] \qquad (10)$$

where $F_{t,opt}$ is the optimal slip force or frictional force, $V_i$ is the shear load, and $n_i$ is the number of dampers per direction in the story $i$. The shear load was calculated from the results of the Fast Nonlinear Analysis (FNA).

For the diagonal configuration to the damper-brace assembly, it is clear that the device and the brace are connected in series. The FD stiffness value is considered infinity, so the total stiffness value to integrate while modeling is equal to the brace system to avoid brace buckling, as explained in Equation (11) [55]:

$$k_{bd} = \frac{1}{\left(\frac{1}{k_b}\right) + \left(\frac{1}{k_d}\right)} \overset{k_d \to \infty}{\longrightarrow} k_{bd} = k_b \qquad (11)$$

The same two alternatives, as in the optimal design of VDs, were studied in the optimal design of the FDs (see Figure 8). Table 10 summarizes and compares the results. The authors selected configuration number 1 due to certain criteria: the significant reduction obtained in the longitudinal direction in terms of displacement and base shear. Although alternative 2 provides an important reduction in some buildings in the transversal direction, the alternative that is able to reduce top displacement and base shear values as much as

possible was chosen. Figure 11 shows the schematic configuration during the modeling of FDs with the use of N-link plastic elements.

**Table 10.** Responses of the three investigated buildings for the two alternatives of FDs placement.

| Building Type | Direction | Fundamental Period [s] | | Top Roof Displacement [cm] | | Base Shear [kN] | |
|---|---|---|---|---|---|---|---|
| | | Altern. 1 | Altern. 2 | Altern. 1 | Altern. 2 | Altern. 1 | Altern. 2 |
| Low-rise | Longitudinal | 0.158 | 0.158 | 0.228 | 0.290 | 6971.2 | 7296.0 |
| | Transversal | 0.218 | 0.218 | 0.126 | 0.057 | 1525.1 | 2000.0 |
| Mid-rise | Longitudinal | 0.707 | 0.708 | 1.682 | 1.775 | 14,030.2 | 14,639.4 |
| | Transversal | 0.998 | 0.999 | 1.124 | 0.298 | 3100.0 | 3500.0 |
| High-rise | Longitudinal | 2.016 | 2.016 | 6.915 | 7.787 | 12,373.0 | 13,950.1 |
| | Transversal | 2.239 | 2.239 | 4.349 | 2.672 | 3494.58 | 4000.0 |

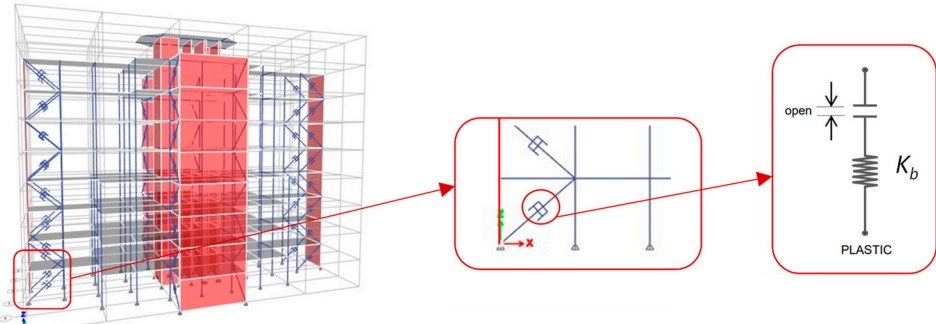

**Figure 11.** Modeling configuration of FDs.

## 5. Results and Discussion

The alternative design buildings with the three passive energy dissipation systems were redesigned. Focus was placed on the optimal design of the dissipated systems. The aim of the attempted redesign was to minimize (i) the maximum displacement at the top of the structures, (ii) the maximum inter-story drift, and (iii) the base shear loads. The results are presented in terms of two essential parameters: the maximum top roof displacement and base shear forces. In addition, the maximum inter-story drift is presented. A comparison of each parameter is established between the undamped and the damped cases with tuned mass damper, viscous, and friction dampers for the seven selected ground motions. An interpretation is established at the end in order to provide a conclusion on the comparative results, and to select the most suitable damper for each type of building.

### 5.1. Displacement at the Top of the Structures

Figure 12 illustrates the horizontal displacement at the top of each building in the longitudinal and transversal direction for the seven accelerograms. The percentage of reduction in the responses for the low-rise building (Figure 12a,b) equipped with friction dampers, in comparison with the structure without dampers, generally exceeds 70.29% in both directions, and reaches 86.10% with ChiChi earthquake excitation in the transversal direction. Although the reduction in the longitudinal direction with viscous dampers is bigger than those obtained with friction dampers, it is limited in the transversal direction, reaching a maximum of 39.49%. As for the damped case with TMD, the percentage of reduction does not exceed 12.94% for all earthquake records except Samos. By evaluating the mean value of percentage reduction in both directions, which is equal to 6.72% and 9.75% with a tuned mass damper, 91.11% and 30.82% with viscous dampers, and 71.44% and 76.87 % with friction dampers, it can be seen that friction dampers perform better than the two other types in the response reduction in the low-rise building.

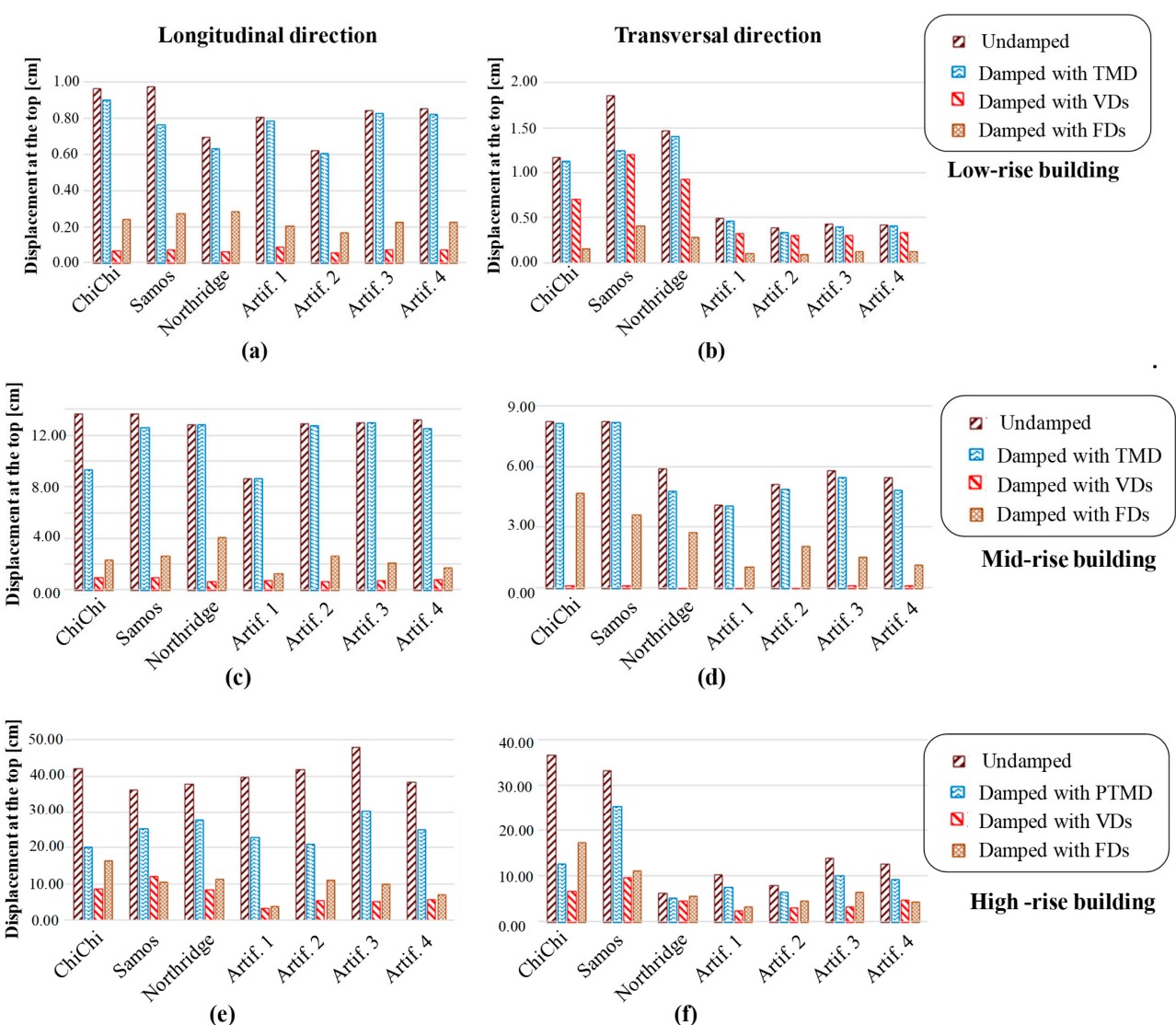

**Figure 12.** Horizontal displacement at the top of (**a**,**b**) the low-rise building, (**c**,**d**) the mid-rise building, and (**e**,**f**) the high-rise building.

The maximum roof displacement value of the mid-rise building for each history record is plotted in Figure 12 with and without dampers. According to the results, it can be seen that utilizing viscous dampers reduces the displacement the most in both horizontal directions, which goes beyond 91.46%. For comparison purposes, the maximum displacement values with friction dampers in an 87.26% and 79.25% reduction and with tuned mass damper in a 31.71% and 19.24% reduction, respectively, in longitudinal and transversal direction. Since the viscous and friction dampers have the greatest impact on the displacement by evaluating the mean value of percentage reduction in both directions, which is equal to 6.58% and 6.30% with a tuned mass damper, 93.70% and 98.96% with viscous dampers, and 81.14% and 63.29% with friction dampers, both systems seem to perform well under all earthquake records for the mid-rise building.

From the results of the high-rise building, all three types of dampers contribute to significant reduction in terms of displacement. One could observe that the percentage of reduction for the high-rise building equipped with friction dampers reaches a maximum of 90.36% in the longitudinal direction, and a maximum of 69.50% in the transversal direction, which is considered high. Viscous dampers also provide high values of reduction, reaching a maximum of 91.89% and 76.96% in both horizontal directions, respectively. Moreover, the pendulum configuration of PTMD offers a great reduction of 52.10% and 65.75% for the

longitudinal and transversal direction, respectively. In terms of mean values, a reduction of 39% and 30% is detected with a tuned mass damper, 82.53% and 65.71% with viscous dampers, and 75.28% and 51.75% with friction dampers. Considering the mean values listed before, even though the PTMD system performs less than the two other damping systems, the reduction results are considered acceptable.

*5.2. Base Shear Load*

Figure 13 shows the results of the base shear load under the seven seismic ground records for the undamped and the three damped cases. For all damper systems, the ratio between the base shears of models with and without dampers for the low-rise building shows an important reduction (Figure 13a,b). Up to a 76.74% response reduction was achieved with viscous dampers in the longitudinal direction, and a maximum reduction attained by 36.61% in transversal direction. High values of reduction with friction dampers reach 39.93% and 59.14%, respectively, in both directions. With TMD, the percent of reductions are given as about a maximum of 15.74%. By evaluating the mean value of percentage reduction in both directions, which is equal to 2.54% and 5.32% with tuned mass damper, 79.96% and 31.06% with viscous dampers, and 27.38% and 41.66 % with friction dampers, both viscous and friction dampers appear to be effective for the low-rise building in base shear reduction.

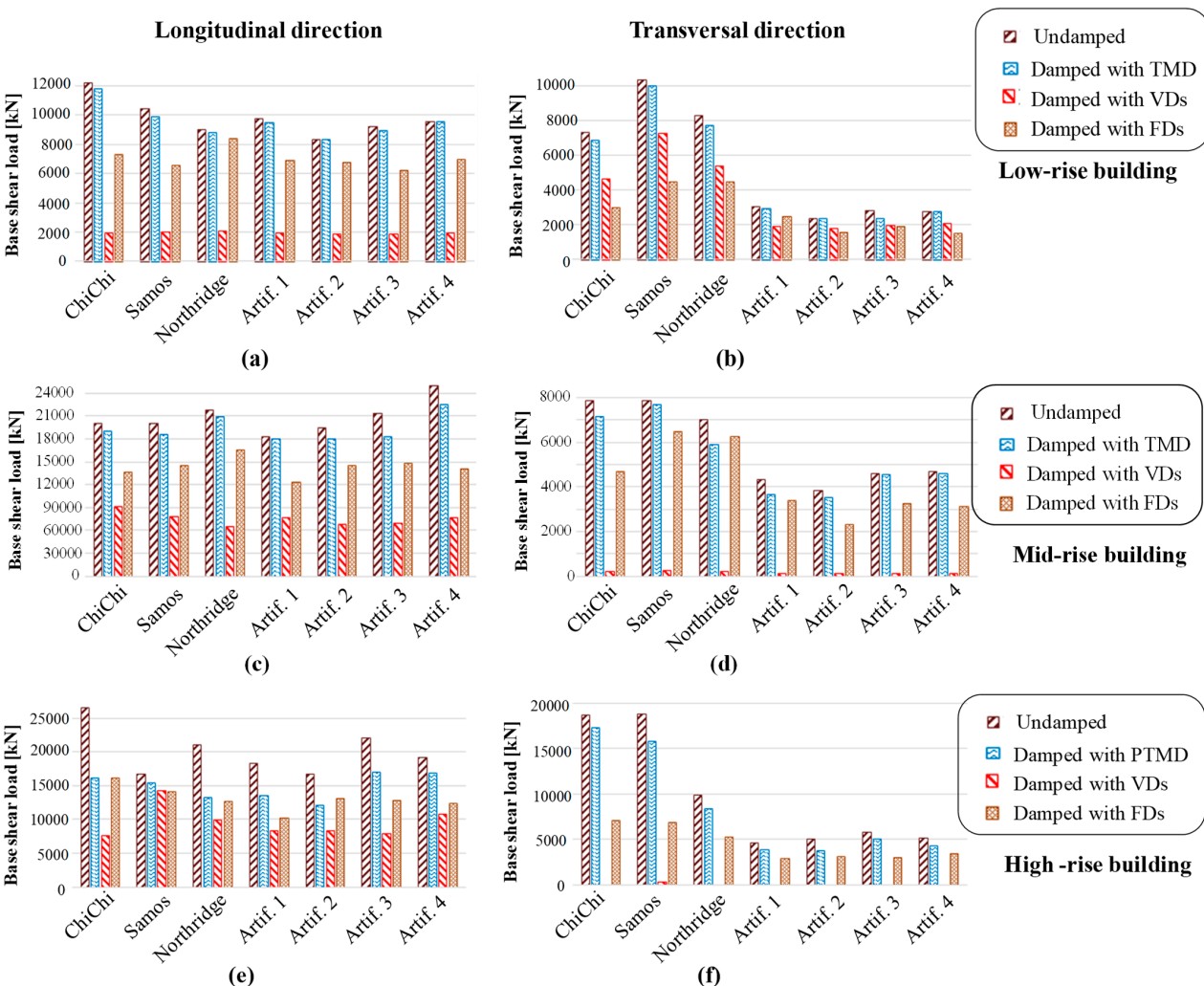

**Figure 13.** Base shear load for (**a**,**b**) the low-rise building, (**c**,**d**) the mid-rise building and (**e**,**f**) the high-rise building.

Tuning of the three studied dampers resulted in a 14.79%, 70.09%, and 43.59% maximum reduction of the base shear values with the tuned mass damper, viscous dampers, and friction dampers systems, respectively, in the longitudinal direction, and in a 16.14%, 98.22%, and 40.70% maximum reduction, respectively, in the transversal direction. Considering all the results obtained, and by evaluating the mean value of percentage reduction in both directions, which is equal to 7.12% and 7.72% with tuned mass damper, 63.91% and 97.62% with viscous dampers, and 30.97% and 27.57% with friction dampers, the mid-rise building equipped with viscous dampers have the greatest impact on the reduction of the base shear response in earthquake excitations compared to the two other damping systems.

Base shear results indicated a maximum reduction achieved with viscous dampers reaching 71.20% and 98.76% for ChiChi earthquake in longitudinal and transversal direction, respectively, and a maximum of 43.77% and 63.48% with friction dampers.

### 5.3. Interstory Drift

Due to a large number of diagrams, the authors have decided to present the diagrams of the interstory drift only for the accelerogram Samos (Figures 14 and 15). The interstory drift index is defined as interstory displacement, $\delta_{s,i}$, divided by story height, $h_i$. The relationship between the interstory drift index and the global drift index $\delta_t/h_t$ depends on the extent of inelasticity in the structure, the type of plastic hinge mechanism, and the importance of higher mode effects. This comparison validates the general conclusion of this study that is presented in the Section 6.

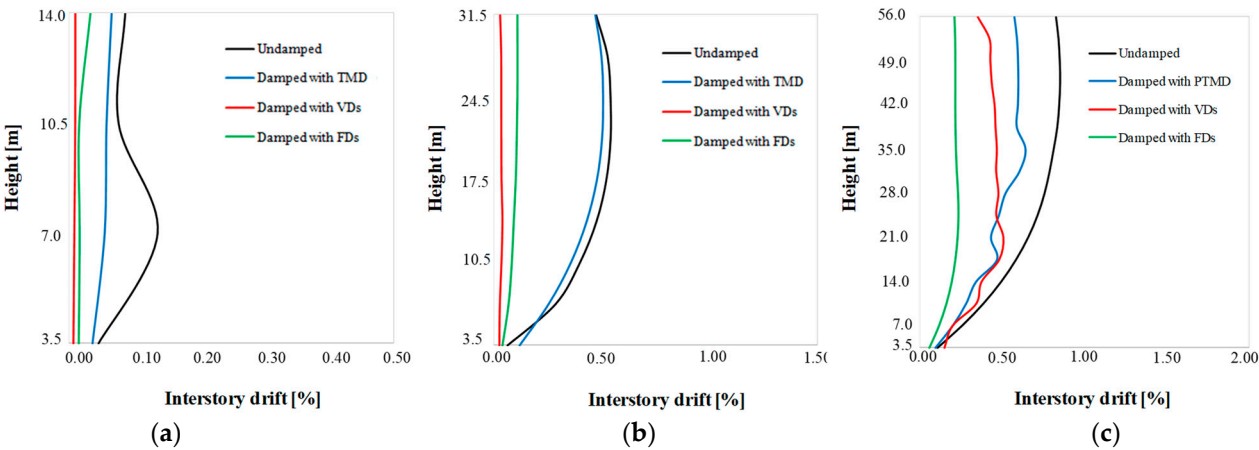

**Figure 14.** Maximum interstory drift for (**a**) low-rise, (**b**) mid-rise, and (**c**) high-rise building in the longitudinal direction.

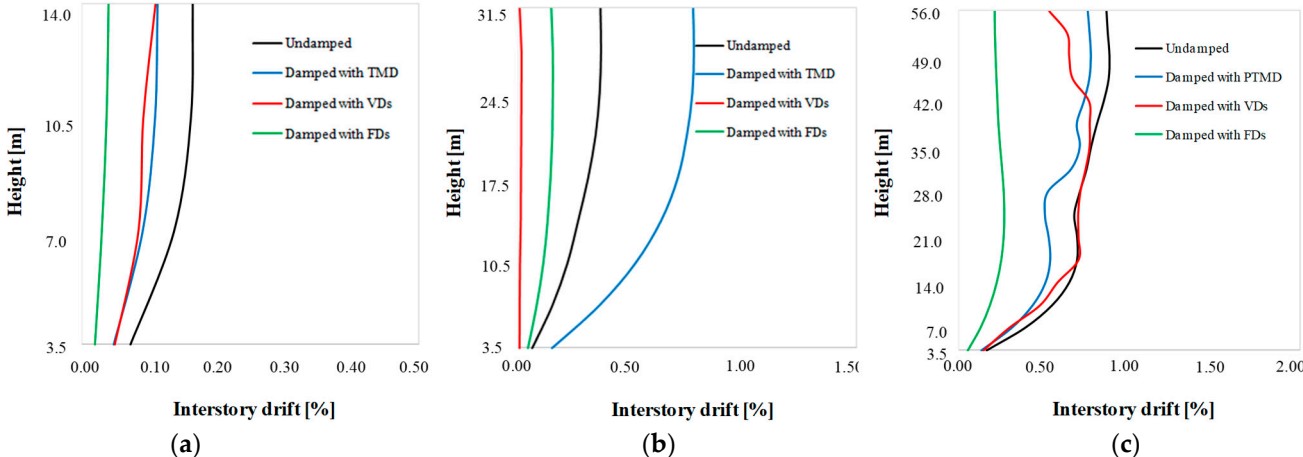

**Figure 15.** Maximum interstory drift for (**a**) low-rise, (**b**) mid-rise, and (**c**) high-rise building in the transversal direction.

### 5.4. Hysteretic Loops

In this section of our study, the hysteresis loops of the three dampers (PTMD, VD and FD) are presented in Figure 16. This figure presents the loops of the real accelerogram of Samos for the high-rise building. The shape of the loops is compared with the expected and well known shape of each damper based on previous studies [1–3,15–17,21–24,56,57] and the accuracy of this study is qualitative.

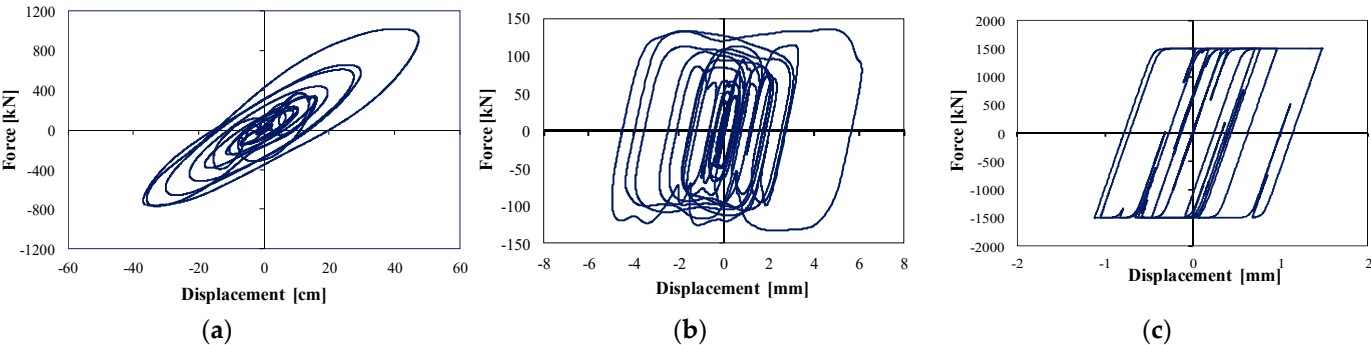

**Figure 16.** Hysteretic loops for (**a**) PTMD, (**b**) VD, (**c**) FD.

### 5.5. General Remarks

The comparative evaluation of the results obtained between undamped and damped case, discussed in terms of displacement and base shear, has led to the following interpretations:

- It is well known that the structural response reduction increases as the mass of TMD increases, but this mass has a limit in practice, due to geometrical and economical constraints. That is the reason why the mass ratio is not considered as an important value to optimize, and therefore, it is hard to achieve high reduction values practically. The results show that TMD systems are not effective for low and mid-rise buildings, because both the displacement and base shear values are barely affected, unlike high-rise building values. In fact, TMD are motion-based systems that demonstrate how their effectiveness is very limited for rigid buildings. As for the high-rise building, even though damped case with PTMD provides less reduction compared to the two other damped cases, it is considered acceptable and more suitable for this kind of structure.
- Structural strengthening with viscous damper systems is defined by the desired additional damping fixed in the preliminary design. From the results obtained, it has been observed that the structural response with the viscus dampers decreases well, showing better performance in terms of the displacement and base shear. In addition, viscous dampers are velocity-dependent systems, where its effectiveness increases with high velocities, usually for flexible buildings. Even though these systems are considered effective for the three studied buildings, they are considered more suitable for mid-rise buildings.
- Friction dampers' incorporation into the structures reduces considerably the building's response after optimizing dampers slip forces, their numbers, and locations under all earthquakes and types of buildings considered. It can be seen from the results obtained that the friction dampers are effective for both rigid and flexible buildings.

## 6. Conclusions

The present study compares the seismic response of three reinforced concrete (RC) symmetric buildings with varied number of stories strengthening with three types of passive energy dissipation systems, as tuned mass dampers, viscous dampers, and friction dampers. We focus the optimal design of each building on minimizing (i) the maximum displacement at the top of the structures, (ii) the base shear loads and (iii) the maximum interstory drift. Three residents' buildings (a four-story building, a nine-story building,

and a sixteen-story building) were subjected to seven (real and artificial) seismic recorded accelerograms. The buildings were tested by considering a nonlinear dynamic analysis. The selected recorded time history functions fulfill the spectrum compatibility conditions required by the Eurocode.

The objective of the present paper was to optimize damper properties and placement in the selected buildings in order to maximize structural performance by providing high reduction, especially in terms of the displacement and base shear. A comparison was established between obtained results with the three types of passive dampers used, including tuned mass dampers (TMD), viscous dampers (VD), and friction dampers (FD) to choose the most suitable damping system for each type of structure, taking into account the damping–cost general relationship.

As a conclusion, the friction dampers were found suitable for the low-rise building, the viscous damping more preferable to incorporate in the mid-rise building, and the pendulum configuration of the tuned mass damper system more appropriate for the high-rise building. These results are valid for the previous symmetric structures under the earthquake considered in the present study. However, they provide a good insight into the effect of strengthening solutions with passive energy dissipation systems in symmetric reinforced concrete buildings. It is important to notice that for further investigation, buildings with other characteristics, such as irregularities in plan and elevation, should be also examined in order to study their effect on dampers design optimization, and choosing the most appropriate strengthening solutions for irregular buildings with different heights.

**Author Contributions:** All the authors contributed to the design and implementation of the research, to the analysis of the results and to the writing of the manuscript. All authors have read and agreed to the published version of the manuscript.

**Funding:** This research received no external funding.

**Institutional Review Board Statement:** Not applicable.

**Informed Consent Statement:** Not applicable.

**Data Availability Statement:** Not applicable.

**Conflicts of Interest:** The authors declare no conflict of interest.

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
