# Peer review of "Comparison of Strengthening Solutions with Optimized Passive Energy Dissipation Systems in Symmetric Buildings"

_applsci, doi:10.3390/app112110103_

Round 1

Reviewer 1 Report

Brief Description

In this study, nonlinear response history analyzes were performed to find the optimal strengthening options for several building cases. The building cases are a low-rise building (four-story building), a mid-rise building (nine-story building), and a high-rise building (sixteen-story building). Nonlinear response history analyzes using seven (real and artificial) seismic recorded accelerograms is conducted to determine which energy dissipation system is appropriate for each case, between tuned mass dampers, viscous dampers, or friction dampers. Parameters used to assess structural performance are displacement at the top of the structures and maximum inter-story drift. From this study, it was found that the use of tuned mass dampers is good for high-rise buildings, viscous dampers are good for mid-rise buildings, and friction dampers are good for low-rise buildings.

General Comment / Review

the study conducted is considered too narrow by only using two possible values for TMD and two possible alternatives for viscous and friction dampers. The focus of the study is quite disperse because initially, it was stated that structural performance was only assessed from top roof displacement and maximum inter-story drift, but in this study, base shear and cost-related parameters are also considered.

Detailed Comments / Reviews

Row Number / Section

Comments

88

The definition of 'regular in plan' needs to be clarified because when referring to the general code of irregularity, the building has a reentrant corner irregularity (source: Table 12.3-1 ASCE 7-16).

Table 1

If the building is not a real building, and the building data is assumed by the author(s), the use of beam dimensions of 30x30 cm is considered relatively small. The use of 30x80 cm columns in mid-rise and high-rise buildings also does not meet the general provisions for special moment frame columns (source: Provision 18.7.2.1 ACI 318-19).

103

The assumption of using a rigid floor diaphragm must be strengthened by proving that the building does not have horizontal irregularities. The use of semi-rigid is recommended to make the distribution of earthquake forces in the structural diaphragm more realistic.

229

The trial process using two values is too limited.

Table 5 & 6

The study aims to find the optimal strengthening method in terms of top roof displacement, so for low-rise buildings, it is more optimal for the low-rise building. However, if the authors want to consider the base shear, the use is more optimal for the high-rise building.

Table 9

Regarding the use of the Maxwell model as in the state in row 117, the use of a viscous damper in a building should not change the structural period of the building, but in this table, it can be seen that the structural periods have changed.

Table 9 & 10

It is better to use a format similar to Table 5 because margin/structural improvement is used in further discussion.

396

It is necessary to explain from where the shear load is obtained or calculated (e.g., from SRSS modal response spectrum analysis).

404

Alternative one was chosen only based on observations through the reduction of top roof displacement in the longitudinal direction. In contrast, when viewed in the transverse direction, the reduction is much larger for alternative 2.

Results and discussion

NLRHA uses several ground motions to determine the average structural response from the analysis of these ground motions. The purpose is to get the expected structural response under the elastic design spectrum. The results of the structural response of each ground motion are less effective to discuss and are not the main goal of conducting NLRHA.

496

As explained above, it is unnecessary to display all analysis results. Instead, the average results of all analyses are preferred to be discussed.

499

The global drift index is not used or shown in the next discussion.

533

Strengthening cost-related parameters are required to be calculated or showed if it is used for consideration. It is recommended to stay focused on the main objective of this study to judge the structural performance based on the top roof displacement and story drift only.

537

It is not strongly recommended to limit ground motions selection based on the building site due to the uncertainty of earthquake characteristics.

561

Price/cost/economic-related parameters need to be explained or calculated if the authors want to be taken into consideration in choosing a strengthening solution.

Author Response

First of all, the authors would like to thank the reviewer for the time he/she spent and his/her valuable comments on this article, helping us to improve the quality of this article. The reponses are provided in the attached file. 

Reviewer 2 Report

Line 29: „…which results in some of this energy…“ – what is “this” energy?

Line 33: What is “supporting structure of the construction”?

In paragraph 2 it would be beneficial to describe in more detail the structural system for resisting horizontal loads (frame, wall, mixed frame-wall…)

Line 103: what kind of “shell” elements were used? How many and which DOF? Also, how was a rigid floor diaphragm modelled? Please explain what kind of inter story structure is present in these buildings, and thus why it is possible to use rigid floor diaphragm model?

Line 105: Is the stiffness of ALL elements reduced by half? Please explain in more detail how and why is this assumption realized?

Line 115: What is the difference between “mass” and “weight” in this context?

Line 125: What is the “frame” element?

Please give a graphical representation of the models used for TMD, VD and FD as explained in paragraph 3, and their position in your structural model.

Please add a reference to table 4 according to which these parameters were calculated.

There are some overlapping numbers in Table 6.

Author Response

First of all, we thank the reviewer for the time he/she spent and his/her valuable comments on our article. The reponses are provided in the attached file. 
